# Partial Information Decomposition: Redundancy as Information Bottleneck

**DOI:** 10.3390/e26070546

**Published:** 2024-06-26

**Authors:** Artemy Kolchinsky

**Affiliations:** 1ICREA-Complex Systems Lab, Universitat Pompeu Fabra, 08003 Barcelona, Spain; artemyk@gmail.com; 2Universal Biology Institute, The University of Tokyo, Tokyo 113-0033, Japan

**Keywords:** partial information decomposition, information bottleneck, rate distortion, redundancy

## Abstract

The partial information decomposition (PID) aims to quantify the amount of redundant information that a set of sources provides about a target. Here, we show that this goal can be formulated as a type of information bottleneck (IB) problem, termed the “redundancy bottleneck” (RB). The RB formalizes a tradeoff between prediction and compression: it extracts information from the sources that best predict the target, without revealing which source provided the information. It can be understood as a generalization of “Blackwell redundancy”, which we previously proposed as a principled measure of PID redundancy. The “RB curve” quantifies the prediction–compression tradeoff at multiple scales. This curve can also be quantified for individual sources, allowing subsets of redundant sources to be identified without combinatorial optimization. We provide an efficient iterative algorithm for computing the RB curve.

## 1. Introduction

Many research fields that study complex systems are faced with multivariate probabilistic models and high-dimensional datasets. Prototypical examples include brain imaging data in neuroscience, gene expression data in biology, and neural networks in machine learning. In response, various information-theoretic frameworks have been developed in order to study multivariate systems in a universal manner. Here, we focus on two such frameworks, *partial information decomposition* and the *information bottleneck*.

The *partial information decomposition* (PID) considers how information about a target random variable *Y* is distributed among a set of source random variables $X_{1},\ldots ,X_{n}$ [1,2,3,4]. For example, in neuroscience, the sources $X_{1},\ldots ,X_{n}$ might represent the activity of *n* different brain regions and *Y* might represent a stimulus, and one may wish to understand how information about the stimulus is encoded in different brain regions. A central idea of the PID is that the information provided by the sources can exhibit *redundancy*, when the same information about *Y* is present in each source, and *synergy*, when information about *Y* is found only in the collective outcome of all sources. Moreover, it has been shown that standard information-theoretic quantities, such as entropy and mutual information, are not sufficient to quantify redundancy and synergy [1,5]. However, finding the right measures of redundancy and synergy has proven difficult. In recent work [4], we showed that such measures can be naturally defined by formalizing the analogy between set theory and information theory that lies at the heart of the PID [5]. We then proposed a measure of redundant information (*Blackwell redundancy*) that is motivated by algebraic, axiomatic, and operational considerations. We argued that Blackwell redundancy overcomes many limitations of previous proposals [4].

The *information bottleneck* (IB) [6,7] is a method for extracting compressed information from one random variable *X* that optimally predicts another target random variable *Y*. For instance, in the neuroscience example with stimulus *Y* and brain activity *X*, the IB method could be used to quantify how well the stimulus can be predicted using only one bit of information about brain activity. The overall tradeoff between the prediction of *Y* and compression of *X* is captured by the so-called *IB curve*. The IB method has been employed in various domains, including neuroscience [8], biology [9], and cognitive science [10]. In recent times, it has become particularly popular in machine learning applications [7,11,12,13,14].

In this paper, we demonstrate a formal connection between PID and IB. We focus in particular on the relationship between the IB and PID redundancy, leaving the connection to other PID measures (such as synergy) for future work. To begin, we show that Blackwell redundancy can be formulated as an information-theoretic constrained optimization problem. This optimization problem extracts information from the sources that best predict the target, under the constraint that the solution does not reveal which source provided the information. We then define a generalized measure of Blackwell redundancy by relaxing the constraint. Specifically, we ask how much predictive information can be extracted from the sources without revealing more than a certain number of bits about the identity of the source. Our generalization leads to an IB-type tradeoff between the prediction of the target (generalized redundancy) and compression (leakage of information about the identity of the source). We refer to the resulting optimization problem as the *redundancy bottleneck* (RB) and to the manifold of optimal solutions at different points on the prediction/compression tradeoff as the *RB curve*. We also show that the RB prediction and compression terms can be decomposed into contributions from individual sources, giving rise to an individual RB curve for each source.

Besides the intrinsic theoretical interest of unifying PID and the IB, our approach brings about several practical advantages. In particular, the RB curve offers a fine-grained analysis on PID redundancy, showing how redundant information emerges at various scales and across different sources. This fine-grained analysis can be used to uncover sets of redundant sources without performing intractable combinatorial optimization. Our approach also has numerical advantages. The original formulation of Blackwell redundancy was based on a difficult optimization problem that becomes infeasible for larger systems. By reformulating Blackwell redundancy as an IB-type problem, we are able to solve it efficiently using an iterative algorithm, even for larger systems (code available at https://github.com/artemyk/pid-as-ib, accessed on 12 May 2024). Finally, the RB has some attractive formal properties. For instance, unlike the original Blackwell redundancy, the RB curve is continuous in the underlying probability distributions.

This paper is organized as follows. In the next section, we provide the background on the IB, PID, and Blackwell redundancy. In Section 3, we introduce the RB, illustrate it with several examples, and discuss its formal properties. In Section 4, we introduce an iterative algorithm to solve the RB optimization problem. We discuss the implications and possible future directions in Section 5. All proofs are found in the Appendix A.

## 2. Background

We begin by providing relevant background on the information bottleneck, partial information decomposition, and Blackwell redundancy.

### 2.1. Information Bottleneck (IB)

The information bottleneck (IB) method provides a way to extract information that is present in one random variable *X* that is relevant for predicting another target random variable *Y* [6,15,16]. To do so, the IB posits a “bottleneck variable” *Q* that obeys the Markov condition Q−X−Y. This Markov condition guarantees that *Q* does not contain any information about *Y* that is not found in *X*. The quality of any particular choice of bottleneck variable *Q* is quantified via two mutual information terms: I(X;Q), which decreases when *Q* provides a more compressed representation of *X*, and I(Y;Q), which increases when *Q* allows a better prediction of *Y*. The IB method selects *Q* to maximize prediction given a constraint on compression [15,16,17]:(1)$$I_{\mathrm{IB}}\left(R\right)=\max_{Q:Q-X-Y}I\left(Y;Q\right)\;\mathrm{where}\;I\left(X;Q\right)\leq R.$$
The values of $I_{\mathrm{IB}}\left(R\right)$ for different *R* specify the *IB curve*, which encodes the overall tradeoff between prediction and compression.

In practice, the IB curve is usually explored by considering the Lagrangian relaxation of the constrained optimization problem (Equation 1):(2)$$F_{\mathrm{IB}}\left(\beta \right):=\max_{Q}I\left(Y;Q\right)-\frac{1}{\beta }I\left(X;Q\right)$$
Here, β≥0 is a parameter that controls the tradeoff between compression cost (favored for β→0) and prediction benefit (favored for β→∞). The advantage of the Lagrangian formulation is that it avoids the non-linear constraint in Equation (Equation 1). If the IB curve is strictly concave, then the two Equations (Equation 1) and (Equation 2) are equivalent, meaning that there is a one-to-one map between the solutions of both problems [18]. When the IB curve is not strictly concave, a modified objective such as the “squared Lagrangian” or “exponential Lagrangian” should be used instead; see Refs. [18,19,20] for more details.

Since the original proposal, many reformulations, generalizations, and variants of the IB have been developed [7]. Notable examples include the “conditional entropy bottleneck” (CEB) [13,21], the “multi-view IB” [22], the “distributed IB” [23], as well as a large family of objectives called the “multivariate IB” [24]. All of these approaches consider some tradeoff between two information-theoretic terms: one that quantifies the prediction of target information that should be maximized and one that quantifies the compression of unwanted information that should be minimized. We refer to an optimization that involves a tradeoff between information-theoretic prediction and compression terms as an *IB-type problem*.

### 2.2. Partial Information Decomposition

The PID considers how information about a target random variable *Y* is distributed across a set of source random variables $X_{1},\ldots ,X_{n}$. One of the main goals of the PID is to quantify redundancy, the amount of shared information that is found in each of the individual sources. The notion of redundancy in PID was inspired by an analogy between sets and information that has re-appeared in various forms throughout the history of information theory [25,26,27,28,29,30,31]. Specifically, if the amount of information provided by each source is conceptualized as the size of a set, then the redundancy is conceptualized as the size of the intersection of those sets [1,4,5]. Until recently, however, this analogy was treated mostly as an informal source of intuition, rather than a formal methodology.

In a recent paper [4], we demonstrated that the terms of PID can be defined by formalizing this analogy to set theory. Recall that, in set theory, the intersection of sets $A_{1},\ldots ,A_{n}$ is defined as the largest set *B* that is contained in each set $A_{s}$ for s∈{1…n}. Thus, the size of the intersection of finite sets $A_{1},\ldots ,A_{n}$ is
$$\left|\overset{n}{\underset{s=1}{⋂}}A_{s}\right|=\max_{B}\left|B\right|\;\mathrm{where}\;B\subseteq A_{s}\;\forall s\in \left\{1\ldots n\right\}.$$
We showed that PID redundancy can be defined in a similar way: the redundancy between sources $X_{1},\ldots ,X_{n}$ is the maximum mutual information in any random variable *Q* that is less informative about the target *Y* than each individual source [4]:(3)$$I_{\cap }^{⊑}:=\max_{Q}I\left(Q;Y\right)\;\mathrm{where}\;Q⊑X_{s}\;\forall s\in \left\{1\ldots n\right\}.$$
The notation $Q⊑X_{s}$ indicates that *Q* is “less informative” about the target than $X_{s}$, given some pre-specified ordering relation ⊑. The choice of the ordering relation completely determines the resulting redundancy measure $I_{\cap }^{⊑}$. We discuss possible choices in the following subsection.

We used a similar approach to define “union information”, which in turn leads to a principled measure of synergy [4]. Note that union information and redundancy are related algebraically but not numerically; in particular, unlike in set theory, the principle of inclusion–exclusion does not always hold.

As mentioned above, here, we focus entirely on redundancy and leave the exploration of connections between IB and union information/synergy for future work.

### 2.3. Blackwell Redundancy

Our definition of PID redundancy (Equation 3) depends on the definition of the “less informative” relation ⊑. Although there are many relations that can be considered [25,32,33,34,35], arguably the most natural choice is the *Blackwell order*.

The Blackwell order is a preorder relation over “channels”, that is conditional distributions with full support. A channel $\kappa_{B|Y}$ is said to be less informative than $\kappa_{C|Y}$ in the sense of the Blackwell order if there exists some other channel $\kappa_{B|C}$ such that
(4)$$\begin{array}{cc} \kappa_{B|Y} & =\kappa_{B|C}\circ \kappa_{C|Y}. \end{array}$$
Throughout, we use the notation ∘ to indicate the composition of channels, as defined via matrix multiplication. For instance, $\kappa_{B|Y}=\kappa_{B|C}\circ \kappa_{C|Y}$ is equivalent to the statement $\kappa_{B|Y}\left(b|y\right)=\sum_{c}\kappa_{B|C}\left(b|c\right)\kappa_{C|Y}\left(c|y\right)$ for all *b* and *y*. Equation (Equation 4) implies that $\kappa_{B|Y}$ is less informative than $\kappa_{C|Y}$ if $\kappa_{B|Y}$ can be produced by downstream stochastic processing of the output of channel $\kappa_{C|Y}$. We use the notation
(5)$$\kappa_{B|Y}⪯\kappa_{C|Y},$$
to indicate that $\kappa_{B|Y}$ is less Blackwell-informative than $\kappa_{C|Y}$. The Blackwell order can also be defined over random variables rather than channels. Given a target random variable *Y* with full support, random variable *B* is said to be less Blackwell-informative than random variable *C*, written as
(6)$$B{⪯}_{Y}C,$$
when their corresponding conditional distributions obey the Blackwell relation, $p_{B|Y}⪯p_{C|Y}$ [36]. It is not hard to verify that any random variable *B* that is independent of *Y* is lowest under the Blackwell order, obeying $B{⪯}_{Y}C$ for all *C*.

The Blackwell order plays a fundamental role in statistics, and it has an important operational characterization in decision theory [36,37,38]. Specifically, $p_{B|Y}⪯p_{C|Y}$ if and only if access to channel $p_{C|Y}$ is better for every decision problem than access to channel $p_{B|Y}$. See Refs. [4,39] for details of this operational characterization and Refs. [4,36,39,40,41,42] for more discussion of the relation between the Blackwell order and the PID.

Combining the Blackwell order (Equation 6) with Equation (Equation 3) gives rise to *Blackwell redundancy* [4]. Blackwell redundancy, indicated here as $I_{\cap }$, is the maximal mutual information in any random variable that is less Blackwell-informative than each of the sources:(7)$$I_{\cap }:=\max_{Q}\;I\left(Q;Y\right)\;\mathrm{where}\;Q{⪯}_{Y}X_{s}\;\forall s.$$
The optimization is always well defined because the feasible set is not empty, given that any random variable *Q* that is independent of *Y* satisfies the constraints. (Note also that, for continuous-valued or countably infinite sources, max may need to be replaced by a sup; see also Appendix A.)

$I_{\cap }$ has many attractive features as a measure of PID redundancy, and it overcomes several problems with previous approaches [4]. For instance, it can be defined for any number of sources, it uniquely satisfies a natural set of PID axioms, and it has fundamental statistical and operational interpretations. Statistically, it is the maximum information transmitted across any channel that can be produced by downstream processing of any one of the sources. Operationally, it is the maximum information that any random variable can have about *Y* without being able to perform better on any decision problem than any one of the sources.

As we showed [4], the optimization problem (Equation 7) can be formulated as the maximization of a convex function subject to a set of linear constraints. For a finite-dimensional system, the feasible set is a finite-dimensional polytope, and the maximum will lie on one of its extreme points; therefore, the optimization can be solved exactly by enumerating the vertices of the feasible set and choosing the best one [4]. However, this approach is limited to small systems, because the number of vertices of the feasible set can grow exponentially.

Finally, it may be argued that Blackwell redundancy is actually a measure of redundancy in the channels $p_{X_{1}|Y},\ldots ,p_{X_{n}|Y}$, rather than in the random variables $X_{1},\ldots ,X_{n}$. This is because the joint distribution over $\left(Y,X_{1},\ldots ,X_{n}\right)$ is never explicitly invoked in the definition of $I_{\cap }$; in fact, any joint distribution is permitted as long as it is compatible with the correct marginals. (The same property holds for several other redundancy measures ([4], Table 1), and Ref. [39] even suggested this property as a requirement for any valid measure of PID redundancy.) In some cases, the joint distribution may not even exist, for instance when different sources represent mutually exclusive conditions. To use a neuroscience example, imagine that $p_{X_{1}|Y}$ and $p_{X_{2}|Y}$ represent the activity of some brain region *X* in response to stimulus *Y*, measured either in younger ($p_{X_{1}|Y}$) or older ($p_{X_{2}|Y}$) subjects. Even though there is no joint distribution over $\left(Y,X_{1},X_{2}\right)$ in this case, redundancy is still meaningful as the information about the stimulus that can be extracted from the brain activity of either age group. In the rest of this paper, we generally work within the channel-based interpretation of Blackwell redundancy.

## 3. Redundancy Bottleneck

In this section, we introduce the redundancy bottleneck (RB) and illustrate it with examples. Generally, we assume that we are provided with the marginal distribution $p_{Y}$ of the target random variable *Y*, as well as *n* source channels $p_{X_{1}|Y},\ldots ,p_{X_{n}|Y}$. Without loss of generality, we assume that $p_{Y}$ has full support. We use calligraphic letters (like Y and $X_{s}$) to indicate the set of outcomes of random variables (like *Y* and $X_{s}$). For simplicity, we use notation appropriate for discrete-valued variables, such as in Equation (Equation 4), though most of our results also apply to continuous-valued variables.

### 3.1. Reformulation of Blackwell Redundancy

We first reformulate Blackwell redundancy (Equation 7) in terms of a different optimization problem. Our reformulation will make use of the random variable *Y*, along with two additional random variables, *S* and *Z*. The outcomes of *S* are the indexes of the different sources, S={1,…,n}. The set of outcomes of *Z* is the union of the outcomes of the individual sources, $Z={⋃}_{s=1}^{n}X_{s}$. For example, if there are two sources with outcomes $X_{1}=\left\{0,1\right\}$ and $X_{2}=\left\{0,1,2\right\}$, then S={1,2} and Z={0,1}∪{0,1,2}={0,1,2}. The joint probability distribution over (Y,S,Z) is defined as
(8)$$p_{YSZ}\left(y,s,z\right)=\left\{\begin{array}{cc} p_{Y}\left(y\right)\nu_{S}\left(s\right)p_{X_{s}|Y}\left(z|y\right) & \mathrm{if}\;z\in X_{s}\\ 0 & \mathrm{otherwise} \end{array}\right.$$
In other words, *y* is drawn from the marginal $p_{Y}$, the source *s* is then drawn independently from the distribution $\nu_{S}$, and finally *z* is drawn from the channel $p_{X_{s}|Y}\left(z|y\right)$ corresponding to source *s*. In this way, the channels corresponding to the *n* sources ($p_{X_{1}|Y},\ldots ,p_{X_{n}|Y}$) are combined into a single conditional distribution $p_{Z|SY}$.

We treat the distribution $\nu_{S}$ as an arbitrary fixed parameter, and except where otherwise noted, we make no assumptions about this distribution except that it has full support. As we will see, different choices of $\nu_{S}$ cause the different sources to be weighed differently in the computation of the RB. We return to the question of how to determine this distribution below.

Note that, under the distribution defined in Equation (Equation 8), *Y* and *S* are independent, so
(9)I(Y;S)=0.
Actually, many of our results can be generalized to the case where there are correlations between *S* and *Y*. We leave exploration of this generalization for future work.

In addition to *Y*, *Z*, and *S*, we introduce another random variable *Q*. This random variable obeys the Markov condition Q−(Z,S)−Y, which ensures that *Q* does not contain any information about *Y* that is not contained in the joint outcome of *Z* and *S*. The full joint distribution over (Y,S,Z,Q) is
(10)$$p_{YSZQ}\left(y,s,z,q\right)=p_{YSZ}\left(y,s,z\right)p_{Q|SZ}\left(q|s,z\right).$$
We sometimes refer to *Q* as the “bottleneck” random variable.

The set of joint outcomes of (S,Z) with non-zero probability is the disjoint union of the outcomes of the individual sources. For instance, in the example above with $X_{1}=\left\{0,1\right\}$ and $X_{2}=\left\{0,1,2\right\}$, the set of joint outcomes of (S,Z) with non-zero probability is {(1,0),(1,1),(2,0),(2,1),(2,2)}. Because *Q* depends jointly on *S* and *Z*, our results do not depend on the precise labeling of the source outcomes, e.g., they are the same if $X_{2}=\left\{0,1,2\right\}$ is relabeled as $X_{2}=\left\{2,3,4\right\}$.

Our first result shows that Blackwell redundancy can be equivalently expressed as a constrained optimization problem. Here, the optimization is over bottleneck random variables *Q*, i.e., over conditional distributions $p_{Q|SZ}$ in Equation (Equation 10).

**Theorem** **1.**
*Blackwell redundancy (Equation 7) can be expressed as*

(11)
$$\begin{array}{c} I_{\cap }=\max_{Q:Q-(Z,S)-Y}I\left(Q;Y|S\right)\;\mathrm{where}\;I\left(Q;S|Y\right)=0. \end{array}$$



Importantly, Theorem 1 does not depend on the choice of the distribution $\nu_{S}$, as long as it has full support.

In Theorem 1, the Blackwell order constraint in Equation (Equation 7) has been replaced by an information-theoretic constraint I(Q;S|Y)=0, which states that *Q* does not provide any information about the identity of source *S*, additional to that already provided by the target *Y*. The objective I(Q;Y) has been replaced by the conditional mutual information I(Q;Y|S). Actually, the objective can be equivalently written in either form, since I(Q;Y|S)=I(Q;Y) given our assumptions (see the proof of Theorem 1 in the Appendix A). However, the conditional mutual information form will be useful for further generalization and decomposition, as discussed in the next sections.

### 3.2. Redundancy Bottleneck

To relate Blackwell redundancy to the IB, we relax the constraint in Theorem 1 by allowing the leakage of *R* bits of conditional information about the source *S*. This defines the *redundancy bottleneck* (RB) at compression rate *R*: (12)$$I_{\mathrm{RB}}\left(R\right):=\;\max_{Q:Q-(Z,S)-Y}\;I\left(Q;Y|S\right)\;\mathrm{where}\;I\left(Q;S|Y\right)\leq R.$$
We note that, for R>0, the value of $I_{\mathrm{RB}}\left(R\right)$ does depend on the choice of the source distribution $\nu_{S}$.

Equation (Equation 12) is an IB-type problem that involves a tradeoff between prediction I(Q;Y|S) and compression I(Q;S|Y). The prediction term I(Q;Y|S) quantifies the generalized Blackwell redundancy encoded in the bottleneck variable *Q*. The compression term I(Q;S|Y) quantifies the amount of conditional information that the bottleneck variable leaks about the identity of the source. The set of optimal values of (I(Q;S|Y),I(Q;Y|S)) defines the *redundancy bottleneck curve* (RB curve) that encodes the overall tradeoff between prediction and compression.

We prove a few useful facts about the RB, starting from monotonicity and concavity.

**Theorem** **2.**
*$I_{\mathrm{RB}}\left(R\right)$ is non-decreasing and concave as a function of R.*


Since $I_{\mathrm{RB}}\left(R\right)$ is non-decreasing in *R*, the lowest RB value is achieved in the R=0 regime, when it equals the Blackwell redundancy (Theorem 1):(13)$$I_{\mathrm{RB}}\left(R\right)\geq I_{\mathrm{RB}}\left(0\right)=I_{\cap }.$$
The largest value is achieved as R→∞, when the compression constraint vanishes. It can be shown that I(Q;Y|S)≤I(Z;Y|S)=I(Y;Z,S) using the Markov condition Q−(Z,S)−Y and the data-processing inequality (see the next subsection). This upper bound is achieved by the bottleneck variable Q=Z. Combining implies
(14)$$I_{\mathrm{RB}}\left(R\right)\leq I\left(Z;Y|S\right)=\sum_{s}\nu_{S}\left(s\right)I\left(X_{s};Y\right),$$
where we used the form of the distribution $p_{YSZ}$ in Equation (Equation 8) to arrive at the last expression. The range of necessary compression rates can be restricted as 0≤R≤I(Z;S|Y).

Next, we show that, for finite-dimensional sources, it suffices to consider finite-dimensional *Q*. Thus, for finite-dimensional sources, the RB problem (Equation 12) involves the maximization of a continuous objective over a compact domain, so the maximum is always achieved by some *Q*. (Conversely, in the more general case of infinite-dimensional sources, it may be necessary to replace max with sup in Equation (Equation 12); see Appendix A.)

**Theorem** **3.**
*For the optimization problem (Equation 12), it suffices to consider Q of cardinality $\left|Q\right|\leq \sum_{s}\left|X_{s}\right|+1$.*


Interestingly, the cardinality bound for the RB is the same as for the IB if we take X=(Z,S) in Equation (Equation 1) [16,20]. It is larger than the cardinality required for Blackwell redundancy (Equation 7), where $|Q|\leq (\sum_{s}\left|X_{s}\right|)-n+1$ suffices [4].

The Lagrangian relaxation of the constrained RB problem (Equation 12) is given by
(15)$$F_{\mathrm{RB}}\left(\beta \right)=\max_{Q:Q-(Z,S)-Y}I\left(Q;Y|S\right)-\frac{1}{\beta }I\left(Q;S|Y\right).$$
The parameter β controls the tradeoff between prediction and compression. The β→0 limit corresponds to the R=0 regime, in which case, Blackwell redundancy is recovered, while the β→∞ limit corresponds to the R=∞ regime, when the compression constraint is removed. The RB Lagrangian (Equation 15) is often simpler to optimize than the constrained optimization (Equation 12). Moreover, when the RB curve $I_{\mathrm{RB}}\left(R\right)$ is strictly concave, there is a one-to-one relationship between the solutions to the two optimization problems (Equation 12) and (Equation 15). However, when the RB curve is not strictly concave, there is no one-to-one relationship and the usual Lagrangian formulation is insufficient. This can be addressed by optimizing a modified objective that combines prediction and compression in a nonlinear fashion, such as the “exponential Lagrangian” [19]:(16)$$F_{\mathrm{RB}}^{\exp}\left(\beta \right)=\max_{Q:Q-(Z,S)-Y}I\left(Q;Y|S\right)-\frac{1}{\beta }e^{I(Q;S|Y)}.$$
(See an analogous analysis for IB in Refs. [18,19].)

### 3.3. Contributions from Different Sources

Both the RB prediction and compression terms can be decomposed into contributions from different sources, leading to an individual RB curve for each source. As we show in the examples below, this decomposition can be used to identify groups of redundant sources without having to perform intractable combinatorial optimization.

Let *Q* be an optimal bottleneck variable at rate *R*, so that $I_{\mathrm{RB}}\left(R\right)=I\left(Q;Y|S\right)$ and I(Q;S|Y)≤R. Then, the RB prediction term can be expressed as the weighted average of the prediction contributions from individual sources: (17)$$\begin{array}{cc} I_{\mathrm{RB}}\left(R\right)=I\left(Q;Y|S\right) & =\sum_{s}\nu_{S}\left(s\right)I\left(Q;Y|S=s\right). \end{array}$$
Here, we introduce the specific conditional mutual information:(18)$$I\left(Q;Y|S=s\right):=D\left(p_{Q|Y,S=s}∥p_{Q|S=s}\right),$$
where D(·∥·) is the Kullback–Leibler (KL) divergence. To build intuitions about this decomposition, we may use the Markov condition Q−(Z,S)−Y to express the conditional distributions in Equation (Equation 18) as compositions of channels:$$\begin{array}{cc} p_{Q|Y,S=s} & =p_{Q|Z,S=s}\circ p_{Z|Y,S=s}\\ p_{Q|S=s} & =p_{Q|Z,S=s}\circ p_{Z|S=s} \end{array}$$
Using the data-processing inequality for the KL divergence and Equation (Equation 8), we can then write
$$I\left(Q;Y|S=s\right)\leq D\left(p_{Z|Y,S=s}∥p_{Z|S=s}\right)=D\left(p_{X_{s}|Y}∥p_{X_{s}}\right).$$
The last term is simply the mutual information $I(Y;X_{s})$ between the target and source *s*. Thus, the prediction contribution from source *s* is bounded between 0 and the mutual information provided by that source:(19)$$0\leq I\left(Q;Y|S=s\right)\leq I\left(Y;X_{s}\right).$$
The difference between the mutual information and the actual prediction contribution:$$I\left(Y;X_{s}\right)-I\left(Q;Y|S=s\right)\geq 0,$$
quantifies the unique information in source *s*. The upper bound in Equation (Equation 19) is achieved in the R→∞ limit by Q=Z, leading to Equation (Equation 14). Conversely, for R=0, $p_{Q|Y,S=s}=p_{Q|Y}$ (from I(Q;S|Y)=0) and $p_{Q|S=s}=p_{Q}$ (from Equation (Equation 9)), so
$$I\left(Q;Y|S=s\right)=I\left(Q;Y\right)=I_{\mathrm{RB}}\left(0\right)=I_{\cap }.$$
Thus, when R=0, the prediction contribution from each source is the same, and it is equal to the Blackwell redundancy.

The RB compression cost can also be decomposed into contributions from individual sources:(20)$$\begin{array}{cc} I(Q;S|Y) & =\sum_{s}\nu_{S}\left(s\right)I\left(Q;S=s|Y\right). \end{array}$$
Here, we introduce the specific conditional mutual information:(21)$$I\left(Q;S=s|Y\right):=D\left(p_{Q|Y,S=s}∥p_{Q|Y}\right).$$

The source compression terms can be related to so-called *deficiency*, a quantitative generalization of the Blackwell order. Although various versions of deficiency can be defined [43,44,45], here we consider the “weighted deficiency” induced by the KL divergence. For any two channels $p_{B|Y}$ and $p_{C|Y}$, it is defined as
(22)$$\begin{array}{cc} \delta_{D}\left(p_{C|Y},p_{B|Y}\right) & :=\min_{\kappa_{B|C}}D\left(\kappa_{B|C}\circ p_{C|Y}∥p_{B|Y}\right). \end{array}$$
This measure quantifies the degree to which two channels violate the Blackwell order, vanishing when $\kappa_{B|Y}⪯\kappa_{C|Y}$. To relate the source compression terms (Equation 20) to deficiency, observe that $p_{Q|Y,S=s}=p_{Q|Z,S=s}\circ p_{Z|Y,S=s}$ and that $p_{Z|Y,S=s}=p_{X_{s}|Y}$. Given Equation (Equation 21), we then have
(23)$$I\left(Q;S=s|Y\right)\geq \delta_{D}\left(p_{X_{s}|Y},p_{Q|Y}\right).$$
Thus, each source compression term is lower bounded by the deficiency between the source channel $p_{X_{s}|Y}$ and the bottleneck channel $p_{Q|Y}$. Furthermore, the compression constraint in the RB optimization problem (Equation 12) sets an upper bound on the deficiency of $p_{Q|Y}$ averaged across all sources.

Interestingly, several recent papers have studied the relationship between deficiency and PID redundancy in the restricted case of two sources [38,41,45,46,47]. To our knowledge, we provide the first link between deficiency and redundancy for the general case of multiple sources. Note also that previous work considered a slightly different definition of deficiency where the arguments of the KL divergence are reversed. Our definition of deficiency is arguably more natural, since it is more natural to minimize the KL divergence over a convex set with respect to the first argument [48].

Finally, observe that, in both decompositions (Equation 17) and (Equation 20), the source contributions are weighted by the distribution $\nu_{S}\left(s\right)$. Thus, the distribution $\nu_{S}$ determines how different sources play into the tradeoff between prediction and compression. In many cases, $\nu_{S}$ can be chosen as the uniform distribution. However, other choices of $\nu_{S}$ may be more natural in other situations. For example, in a neuroscience context where different sources correspond to different brain regions, $\nu_{S}\left(s\right)$ could represent the proportion of metabolic cost or neural volume assigned to region *s*. Alternatively, when different sources represent mutually exclusive conditions, as in the age group example mentioned at the end of Section 2, $\nu_{S}\left(s\right)$ might represent the frequency of condition *s* found in the data. Finally, it may be possible to set $\nu_{S}$ in an “adversarial” manner so as to maximize the resulting value of $I_{\mathrm{RB}}\left(R\right)$ in Equation (Equation 12). We leave the exploration of this adversarial approach for future work.

### 3.4. Examples

We illustrate our approach using a few examples. For simplicity, in all examples, we use a uniform distribution over the sources, $\nu_{S}\left(s\right)=1/n$. The numerical results are calculated using the iterative algorithm described in the next section.

**Example** **1.**
*We begin by considering a very simple system, called the “UNIQUE gate” in the PID literature. Here, the target *Y* is binary and uniformly-distributed, $p_{Y}\left(y\right)=1/2$ for y∈{0,1}. There are two binary-valued sources, $X_{1}$ and $X_{2}$, where the first source is a copy of the target, $p_{X_{1}|Y}\left(x_{1}|y\right)=\delta_{x_{1},y}$, while the second source is an independent and uniformly-distributed bit, $p_{X_{2}|Y}\left(x_{1}|y\right)=1/2$. Thus, source $X_{1}$ provides 1 bit of information about the target, while $X_{2}$ provides none. The Blackwell redundancy is $I_{\cap }=0$ [4], because it is impossible to extract any information from the sources without revealing that this information came from $X_{1}$.*


We performed RB analysis by optimizing the RB Lagrangian $F_{\mathrm{RB}}\left(\beta \right)$ (Equation 15) at different β. Figure 1a,b show the prediction I(Q;Y|S) and compression I(Q;S|Y) values for the optimal bottleneck variables *Q*. At small β, the prediction converges to the Blackwell redundancy, $I\left(Q;Y|S\right)=I_{\cap }=0$, and there is complete loss of information about source identity, I(Q;S|Y)=0. At larger β, the prediction approaches the maximum $I\left(Q;Y|S\right)=0.5\times I\left(X_{1};Y\right)=0.5\;\mathrm{bit}$, and compression approaches I(Q;S|Y)=I(Z;S|Y)≈0.311bit. Figure 1c shows the RB curve, illustrating the overall tradeoff between prediction and compression.

In the shaded regions of Figure 1a,b, we show the additive contributions to the prediction and compression terms from the individual sources, $\nu_{S}\left(s\right)I\left(Q;Y|S=s\right)$ from Equation (Equation 17) and $\nu_{S}\left(s\right)I\left(Q;S=s|Y\right)$ from Equation (Equation 20), respectively. We also show the resulting RB curves for individual sources in Figure 1d. As expected, only source $X_{1}$ contributes to the prediction at any level of compression.

To summarize, if some information about the identity of the source can be leaked (non-zero compression cost), then improved prediction of the target is possible. At the maximum needed compression cost of 0.311, it is possible to extract 1 bit of predictive information from $X_{1}$ and 0 bits from $X_{2}$, leading to an average of 0.5bits of prediction.

**Example** **2.**
*We now consider the “AND gate”, another well-known system from the PID literature. There are two independent and uniformly distributed binary sources, $X_{1}$ and $X_{2}$. The target *Y* is also binary-valued and determined via $Y=X_{1}\;\mathrm{AND}\;X_{2}$. Then, $p_{Y}\left(0\right)=3/4$ and $p_{Y}\left(1\right)=1/4$, and both sources have the same channel:*

$$p_{X_{s}|Y}\left(x|y\right)=\left\{\begin{array}{cc} 2/3 & \mathrm{if}\;y=0,x=0\\ 1/3 & \mathrm{if}\;y=0,x=1\\ 0 & \mathrm{if}\;y=1,x=0\\ 1 & \mathrm{if}\;y=1,x=1 \end{array}\right.$$

*Because the two source channels are the same, the Blackwell redundancy obeys $I_{\cap }=I\left(Y;X_{1}\right)=I\left(Y;X_{2}\right)=0.311$ bits [4]. From Equations (Equation 13) and (Equation 14), we see that $I_{\mathrm{RB}}\left(R\right)=I_{\cap }$ across all compression rates. In this system, all information provided by the sources is redundant, so there is no strict tradeoff between prediction and compression. The RB curve (not shown) consists of a single point, (I(Q;Y|S),I(Q;S|Y))=(0.311,0).*


**Example** **3.**
*We now consider a more sophisticated example with four sources. The target is binary-valued and uniformly distributed, $p_{Y}\left(y\right)=1/2$ for y∈{0,1}. There are four binary-valued sources, where the conditional distribution of each source s∈{1,2,3,4} is a binary symmetric channel with error probability $\epsilon_{s}$:*

(24)
$$p_{X_{s}|Y}\left(x|y\right)=\left\{\begin{array}{cc} 1-\epsilon_{s} & \mathrm{if}\;y=x\\ \epsilon_{s} & \mathrm{if}\;y\neq x \end{array}\right.$$

*We take $\epsilon_{1}=\epsilon_{2}=0.1$, $\epsilon_{3}=0.2$, and $\epsilon_{4}=0.5$. Thus, sources $X_{1}$ and $X_{2}$ provide most information about the target; $X_{3}$ provides less information; $X_{4}$ is completely independent of the target.*


We performed our RB analysis and plot the RB prediction values in Figure 2a and the compression values in Figure 2b, as found by optimizing the RB Lagrangian at different β. At small β, the prediction converges to the Blackwell redundancy, $I\left(Q;Y|S\right)=I_{\cap }=0$, and there is complete loss of information about source identity, I(Q;S|Y)=0. At large β, the prediction is equal to the maximum I(Z;Y|S)≈0.335bit, and compression is equal to I(Q;S|Y)≈0.104bit. Figure 2c shows the RB curve.

In Figure 2a,b, we show the additive contributions to the prediction and compression terms from the individual sources, $\nu_{S}\left(s\right)I\left(Q;Y|S=s\right)$ and $\nu_{S}\left(s\right)I\left(Q;S=s|Y\right)$, respectively, as shaded regions. We also show the resulting RB curves for individual sources in Figure 2d.

As expected, source $X_{4}$ does not contribute to the prediction at any level of compression, in accord with the fact that $I\left(Q;Y|S=s\right)\leq I\left(X_{4};Y\right)=0$. Sources $X_{1}$ and $X_{2}$ provide the same amount of prediction and compression at all points, up to the maximum $I\left(X_{1};Y\right)=I\left(X_{2};Y\right)\approx 0.531$. Source $X_{3}$ provides the same amount of prediction and compression as sources $X_{1}$ and $X_{2}$, until it hits its maximum prediction $I(X_{3};Y)\approx 0.278$. As shown in Figure 2d, at this point, $X_{3}$ splits off from sources $X_{1}$ and $X_{2}$ and its compression contribution decreases to 0; this is compensated by increasing the compression cost of sources $X_{1}$ and $X_{2}$. The same behavior can also be seen in Figure 2a,b, where we see that the solutions undergo phase transitions as different optimal strategies are uncovered at increasing β. Importantly, by considering the prediction/compression contributions from the the individual sources, we can identify that sources $X_{1}$ and $X_{2}$ provide the most redundant information.

Let us comment on the somewhat surprising fact that, at larger β, the compression cost of $X_{3}$ decreases—even while its prediction contribution remains constant and the prediction contribution from $X_{1}$ and $X_{2}$ increases. At first glance, this appears counter-intuitive if one assumes that, in order to increase prediction from $X_{1}$ and $X_{2}$, the bottleneck channel $p_{Q|Y}$ should approach $p_{X_{1}|Y}=p_{X_{2}|Y}$, thereby increasing the deficiency $\delta_{D}\left(p_{X_{3}|Y},p_{Q|Y}\right)$ and the compression cost of $X_{3}$ via the bound (Equation 23). In fact, this is not the case, because the prediction is quantified via the conditional mutual information I(Q;Y|S), not the mutual information I(Q;Y). Thus, it is possible that the prediction contributions from $X_{1}$ and $X_{2}$ are large, even when the bottleneck channel $p_{Q|Y}$ does not closely resemble $p_{X_{1}|Y}=p_{X_{2}|Y}$.

More generally, this example shows that it is possible for the prediction contribution from a given source to stay the same, or even increase, while its compression cost decreases. In other words, as can be seen from Figure 2d, it is possible for the RB curves of the individual sources to be non-concave and non-monotonic. It is only the overall RB curve, Figure 2c, representing the optimal prediction–compression tradeoff on average, that must be concave and monotonic.

**Example** **4.**
*In our final example, the target consists of three binary spins with a uniform distribution, so $Y=(Y_{1},Y_{2},Y_{3})$ and $p_{Y}\left(y\right)=1/8$ for all *y*. There are three sources, each of which contains two binary spins. Sources $X_{1}$ and $X_{2}$ are both equal to the first two spins of the target *Y*, $X_{1}=X_{2}=\left(Y_{1},Y_{2}\right)$. Source $X_{3}$ is equal to the first and last spin of the target, $X_{3}=\left(Y_{1},Y_{3}\right)$.*


Each source provides $I(Y;X_{s})=2\;\mathrm{bits}$ of mutual information about the target. The Blackwell redundancy $I_{\cap }$ is 1 bit, reflecting the fact that there is a single binary spin that is included in all sources ($Y_{1}$).

We performed our RB analysis and plot the RB prediction values in Figure 3a and the compression values in Figure 3b, as found by optimizing the RB Lagrangian at different β. At small β, the prediction converges to the Blackwell redundancy, $I\left(Q;Y|S\right)=I_{\cap }=1$, and I(Q;S|Y)=0. At large β, the prediction is equal to the maximum I(Z;Y|S)=2bit, and compression is equal to I(Z;S|Y)≈0.459. Figure 3c shows the RB curve. As in the previous example, the RB curve undergoes phase transitions as different optimal strategies are uncovered at different β.

In Figure 3a,b, we show the additive contributions to the prediction and compression terms from the individual sources, $\nu_{S}\left(s\right)I\left(Q;Y|S=s\right)$ and $\nu_{S}\left(s\right)I\left(Q;S=s|Y\right)$, as shaded regions. We also show the resulting RB curves for individual sources in Figure 3d.

Observe that sources $X_{1}$ and $X_{2}$ provide more redundant information at a given level of compression. For instance, as shown in Figure 3d, at source compression I(Q;S=s|Y)≈.25, $X_{1}$ and $X_{2}$ provide 2 bits of prediction, while $X_{3}$ provides only a single bit. This again shows how the RB source decomposition can be used for identifying sources with high levels of redundancy.

### 3.5. Continuity

It is known that the Blackwell redundancy $I_{\cap }$ can be discontinuous as a function of the probability distribution of the target and source channels [4]. In Ref. [4], we explain the origin of this discontinuity in geometric terms and provide sufficient conditions for Blackwell redundancy to be continuous. Nonetheless, the discontinuity of $I_{\cap }$ is sometimes seen as an undesired property.

On the other hand, as we show in this section, the value of RB is continuous in the probability distribution for all R>0.

**Theorem** **4.**
*For finite-dimensional systems and R>0, $I_{\mathrm{RB}}\left(R\right)$ is a continuous function of the probability values $p_{X_{s}|Y}\left(x|y\right)$, $p_{Y}\left(y\right)$, and $\nu_{S}\left(s\right)$.*


Thus, by relaxing the compression constraint in Theorem 1, we “smooth out” the behavior of Blackwell redundancy and arrive at a continuous measure. We illustrate this using a simple example.

**Example** **5.**
*We consider the COPY gate, a standard example in the PID literature. Here, there are two binary-valued sources jointly distributed according to*

$$p_{X_{1}X_{2}}\left(x_{1},x_{2}\right)=\left\{\begin{array}{cc} 1/2-\epsilon /4 & \mathrm{if}\;x_{1}=x_{2}\\ \epsilon /4 & \mathrm{if}\;x_{1}\neq x_{2} \end{array}\right.$$

*The parameter ϵ controls the correlation between the two sources, with perfect correlation at ϵ=0 and complete independence at ϵ=1. The target *Y* is a copy of the joint outcome of the two sources, $Y=(X_{1},X_{2})$.*


It is known that Blackwell redundancy $I_{\cap }$ is discontinuous for this system, jumping from $I_{\cap }=1$ at ϵ=0 to $I_{\cap }=0$ for ϵ>0 [4]. On the other hand, the RB function $I_{\mathrm{RB}}\left(R\right)$ is continuous for R>0. Figure 4 compares the behavior of Blackwell redundancy and RB as a function of ϵ, at R=0.01 bits. In particular, it can be seen that $I_{\mathrm{RB}}\left(R\right)=1$ at ϵ=0 and then decays continuously as ϵ increases.

## 4. Iterative Algorithm

We provide an iterative algorithm to solve the RB optimization problem. This algorithm is conceptually similar to the Blahut–Arimoto algorithm, originally employed for rate distortion problems and later adapted to solve the original IB problem [6]. A Python implementation of our algorithm is available at https://github.com/artemyk/pid-as-ib; there, we also provide updated code to exactly compute Blackwell redundancy (applicable to small systems).

To begin, we consider the RB Lagrangian optimization problem, Equation (Equation 15). We rewrite this optimization problem using the KL divergence:(25)$$F_{\mathrm{RB}}\left(\beta \right)=\max_{r_{Q|SZ}}D\left(r_{Y|QS}∥p_{Y|S}\right)-\frac{1}{\beta }D\left(r_{Q|SY}∥r_{Q|Y}\right).$$
Here, notation like $r_{Q|Y}$, $r_{Y|QS}$, etc., refers to distributions that include *Q* and therefore depend on the optimization variable $r_{Q|SZ}$, while notation like $p_{Y|S}$ refers to distributions that do not depend on *Q* and are not varied under the optimization. Every choice of conditional distribution $r_{Q|SZ}$ induces a joint distribution $r_{YSZQ}=p_{YSZ}r_{Q|SZ}$ via Equation (Equation 10).

We can rewrite the first KL term in Equation (Equation 25) as
$$\begin{array}{cc} D(r_{Y|QS}∥p_{Y|S}) & =D\left(r_{Y|QS}∥p_{Y|S}\right)-\min_{\omega_{YSZQ}}D\left(r_{Y|QS}∥\omega_{Y|QS}\right)\\ \; & =\max_{\omega_{YSZQ}}E_{p_{YSZ}r_{Q|SZ}}\left[\ln\frac{\omega (y|q,s)}{p(y|s)}\right]. \end{array}$$
where E indicates the expectation, and we introduced the variational distribution $\omega_{YSZQ}$. The maximum is achieved by $\omega_{YSZQ}=r_{YSZQ}$, which gives $\omega_{Y|QS}=r_{Y|QS}$. We rewrite the second KL term in Equation (Equation 25) as
$$\begin{array}{cc} D(r_{Q|SY}∥r_{Q|Y}) & =D(r_{Q|SY}r_{Z|SYQ}∥r_{Q|Y}r_{Z|SYQ})\\ \; & =\min_{\omega_{YSZQ}}D\left(r_{Q|SY}r_{Z|SYQ}∥\omega_{Q|Y}\omega_{Z|SYQ}\right). \end{array}$$
Here, we introduce the variational distribution $\omega_{YSZQ}$, where the minimum is achieved by $\omega_{YSZQ}=r_{YSZQ}$. The term $r_{Q|SY}r_{Z|SYQ}$ can be rewritten as
$$\begin{array}{c} r\left(q|s,y\right)r\left(z|s,y,q\right)=\frac{r(z,s,y,q)}{p(s,y)}=\frac{r(q|s,z)p(z,s,y)}{p(s,y)}=r\left(q|s,z\right)p\left(z|s,y\right) \end{array}$$
where we used the Markov condition Q−(S,Z)−Y. In this way, we separate the contribution from the conditional distribution $r_{Q|SZ}$ being optimized.

Combining the above allows us to rewrite Equation (Equation 25) as
(26)$$\begin{array}{c} F_{\mathrm{RB}}\left(\beta \right)=\max_{r_{Q|ZS},\omega_{YSZQ}}E_{p_{YSZ}r_{Q|SZ}}\left[\ln\frac{\omega (y|q,s)}{p(y|s)}\right]-\frac{1}{\beta }D\left(r_{Q|SZ}p_{Z|SY}∥\omega_{Q|Y}\omega_{Z|SYQ}\right). \end{array}$$
We now optimize this objective in an iterative and alternating manner with respect to $r_{Q|SZ}$ and $\omega_{YSZQ}$. Formally, let $L(r_{Q|SZ},\omega_{YSZQ})$ refer to the objective in Equation (Equation 26). Then, starting from some initial guess $r_{Q|SZ}^{\left(0\right)}$, we generate a sequence of solutions
(27)$$\begin{array}{cc} \omega_{YSZQ}^{\left(t+1\right)} & =\underset{\omega_{YSZQ}}{\arg\max}L\left(r_{Q|SZ}^{\left(t\right)},\omega_{YSZQ}\right) \end{array}$$
(28)$$\begin{array}{cc} r_{Q|SZ}^{\left(t+1\right)} & =\underset{r_{Q|SZ}}{\arg\max}L\left(r_{Q|SZ},\omega_{YSZQ}^{\left(t+1\right)}\right) \end{array}$$
Each optimization problem can be solved in closed form. As already mentioned, the optimizer in Equation (Equation 27) is
$$\omega_{YSZQ}^{\left(t+1\right)}=r_{YSZQ}^{\left(t\right)}=r_{Q|SZ}^{\left(t\right)}p_{SZY}.$$
The optimization (Equation 28) can be solved by taking derivatives, giving
$$\begin{array}{c} r^{\left(t+1\right)}\left(q|s,z\right)\propto e^{\sum_{y}p\left(y|s,z\right)\left[\beta \ln\omega^{\left(t\right)}\left(y|q,s\right)-\ln\frac{p(z|s,y)}{\omega^{\left(t\right)}\left(q|y\right)\omega^{\left(t\right)}\left(z|s,y,q\right)}\right]}, \end{array}$$
where the proportionality constant in ∝ is fixed by normalization $\sum_{q}r^{\left(t+1\right)}\left(q|s,z\right)=1$.

Each iteration increases the value of objective L. Since the objective is upper bounded by I(Z;Y|S), the algorithm is guaranteed to converge. However, as in the case of the original IB problem, the objective is not jointly convex in both arguments, so the algorithm may converge to a local maximum or a saddle point, rather than a global maximum. This can be partially alleviated by running the algorithm several times starting from different initial guesses $r_{Q|SZ}^{\left(0\right)}$.

When the RB is not strictly concave, it is more appropriate to optimize the exponential RB Lagrangian (Equation 16) or another objective that combines the prediction and compression terms in a nonlinear manner [18,19]. The algorithm described above can be used with such objectives after a slight modification. For instance, for the exponential RB Lagrangian, we modify (Equation 26) as
(29)$$\begin{array}{c} F_{\mathrm{RB}}^{\exp}\left(\beta \right)=\max_{r_{Q|SZ},\omega_{YSZQ}}E_{p_{YSZ}r_{Q|SZ}}\left[\ln\frac{\omega (y|q,s)}{p(y|s)}\right]-\frac{1}{\beta }e^{D(r_{Q|ZS}p_{Z|SY}∥\omega_{Q|Y}\omega_{Z|SYQ})}. \end{array}$$
A similar analysis as above leads to the following iterative optimization scheme:$$\begin{array}{cc} \omega_{YSZQ}^{\left(t+1\right)} & =r_{Q|SZ}^{\left(t\right)}p_{SZY}\\ r^{\left(t+1\right)}\left(q|s,z\right) & \propto e^{\sum_{y}p\left(y|s,z\right)\left[\beta^{\left(t\right)}\ln\omega^{\left(t\right)}\left(y|q,s\right)-\ln\frac{p(z|s,y)}{\omega^{\left(t\right)}\left(q|y\right)\omega^{\left(t\right)}\left(z|s,y,q\right)}\right]}, \end{array}$$
where $\beta^{\left(t\right)}=\beta e^{-I_{r^{\left(t\right)}}\left(Q;S|Y\right)}$ is an effective inverse temperature. (Observe that, unlike the squared Lagrangian [18], the exponential Lagrangian leads to an effective inverse temperature $\beta^{\left(t\right)}$ that is always finite and converges to β as $I_{r^{\left(t\right)}}\left(Q;S|Y\right)\to 0$.)

When computing an entire RB curve, as in Figure 1a–c, we found good results by annealing, that is by re-using the optimal $r_{Q|SZ}$ found for one β as the initial guess at higher β. For quantifying the value of the RB function $I_{\mathrm{RB}}\left(R\right)$ at a fixed *R*, as in Figure 4, we approximated $I_{\mathrm{RB}}\left(R\right)$ via a linear interpolation of the RB prediction and compression values recovered from the RB Lagrangian at varying β.

## 5. Discussion

In this paper, we propose a generalization of Blackwell redundancy, termed the redundancy bottleneck (RB), formulated as an information-bottleneck-type tradeoff between prediction and compression. We studied some implications of this formulation and proposed an efficient numerical algorithm to solve the RB optimization problem.

We briefly mention some directions for future work.

The first direction concerns our iterative algorithm. The algorithm is only applicable to systems where it is possible to enumerate the outcomes of the joint distribution $p_{QYSZ}$. This is impractical for discrete-valued variables with very many outcomes, as well as continuous-valued variables as commonly found in statistical and machine learning settings. In future work, it would be useful to develop RB algorithms suitable for such datasets, possibly by exploiting the kinds of variational techniques that have recently gained traction in machine learning applications of IB [11,12,13].

The second direction would explore connections between the RB and other information-theoretic objectives for representation learning. To our knowledge, the RB problem is novel to the literature. However, it has some similarities to existing objectives, including among others the conditional entropy bottleneck [13], multi-view IB [22], and the privacy funnel and its variants [49]. Showing formal connections between these objectives would be of theoretical and practical interest, and could lead to new interpretations of the concept of PID redundancy.

Another direction would explore the relationship between RB and information-theoretic measures of causality [50,51]. In particular, if the different sources represent some mutually exclusive conditions—such as the age group example provided at the end of Section 2—then redundancy could serve as a measure of causal information flow that is invariant to the conditioning variable.

Finally, one of the central ideas of this paper is to treat the identity of the source as a random variable in its own right, which allows us to consider what information different bottleneck variables reveal about the source. In this way, we convert the search for topological or combinatorial structure in multivariate systems into an interpretable and differential information-theoretic objective. This technique may be useful in other problems that consider how information is distributed among variables in complex systems, including other PID measures such as synergy [4], information-theoretic measures of modularity [52,53], and measures of higher-order dependency [54,55].

## Figures and Tables

**Figure 1 entropy-26-00546-f001:**
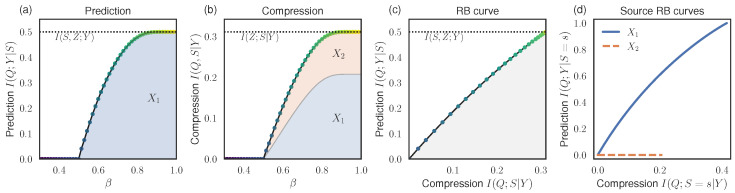
RB analysis for the UNIQUE gate (Example 1). (**a**) Prediction values found by optimizing the RB Lagrangian (Equation 15) at different β. Colored regions indicate contributions from different sources, $\nu_{S}\left(s\right)I\left(Q;Y|S=s\right)$ from Equation (Equation 17). For this system, only source $X_{1}$ contributes to the prediction. (**b**) Compression costs found by optimizing the RB Lagrangian at different β. Colored regions indicate contributions from different sources, $\nu_{S}\left(s\right)I\left(Q;S=s|Y\right)$ from Equation (Equation 20). (**c**) The RB curve shows the tradeoff between optimal compression and the prediction values; the marker colors correspond to the β values as in (**a**,**b**). All bottleneck variables *Q* must fall within the accessible grey region. (**d**) RB curves for individual sources.

**Figure 2 entropy-26-00546-f002:**
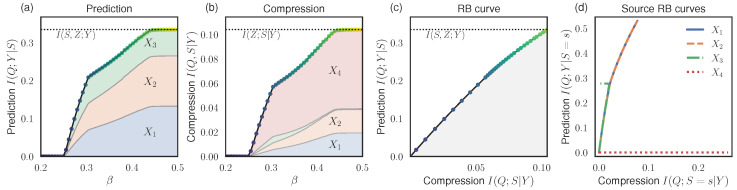
RB analysis for the system with 4 binary symmetric channels (Example 3). (**a**,**b**) Prediction and compression values found by optimizing the RB Lagrangian (Equation 15) at different β. Contributions from individual sources are shown as shaded regions. (**c**) The RB curve shows the tradeoff between optimal compression and prediction values; marker colors correspond to the β values as in (**a**,**b**). (**d**) RB curves for individual sources.

**Figure 3 entropy-26-00546-f003:**
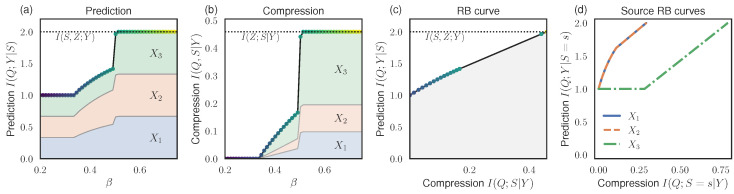
RB analysis for the system with a 3-spin target (Example 4). (**a**,**b**) Prediction and compression values found by optimizing the RB Lagrangian (Equation 15) at different β. Contributions from individual sources are shown as shaded regions. (**c**) The RB curve shows the tradeoff between optimal compression and prediction values; marker colors correspond to the β values as in (**a**,**b**). (**d**) RB curves for individual sources.

**Figure 4 entropy-26-00546-f004:**
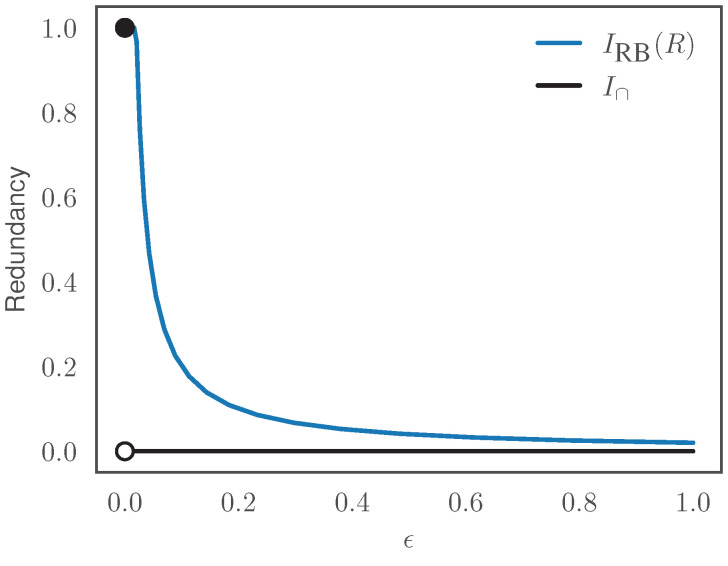
The RB function $I_{\mathrm{RB}}\left(R\right)$ is continuous in the underlying probability distribution for R>0, while Blackwell redundancy can be discontinuous. Here illustrated on the COPY gate, $Y=(X_{1},X_{2})$, as a function of correlation strength ϵ between $X_{1}$ and $X_{2}$ (perfect correlation at ϵ=0, independence at ϵ=1). Blackwell redundancy jumps from $I_{\cap }=1$ at ϵ=0 to $I_{\cap }=0$ at ϵ>0, while $I_{\mathrm{RB}}\left(R\right)$ (at R=0.01) decays continuously.

## Data Availability

No research data was used in this study.

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
