# Peer review of "Partial Information Decomposition: Redundancy as Information Bottleneck"

_entropy, 2024, doi:10.3390/e26070546_

Round 1
Reviewer 1 Report
Comments and Suggestions for Authors
This paper presents a way to frame partial information decomposition as an information bottleneck problem, reframing the Blackwell order PID as a tractable and intuitive variational inference problem. This is a significant development because not only do we get a well-motivated PID measure with desirable properties from the Blackwell order, but we also get a clear and compelling path to practical usage through the information bottleneck formulation. The background and context are well-presented, a practical demonstration is provided, and classic test cases from the literature are analyzed with intuitive outcomes. To me, the only drawback was that the paper doesn't present an example of the IB framework applied at scale using variational methods, but it is reasonable to leave this for future work.
Comments:
- It was surprising to see the joint distribution is not used! Is it fair to say, though, that even though only the conditional distribution p(y|x_s) is used, this somewhat constrains what the joint distribution p(x1, ..., xn) could be? I'm not very familiar with the PID corner cases, but are there any that depend on the joint distribution p(x1, ..., xn)?
- One part that was a little unclear to me is how the domain of Z depends on label identities. For instance, if I say that X1 is in {0,1} and X2 is in {2,3,4}, does it change things? I'm trying to understand how general this part is. My guess is that since Q depends on S and Z, it doesn't matter at all, and the domain union step was just a notational choice. But it would be nice to clarify.
- Is there some natural way to set the measure ν(s)? I could imagine some sort of optimization to set it. Though the choice of a uniform distribution is simple and natural.
- In Eq. 10, there's a typo—missing a variable z.
- You mentioned the privacy funnel. I'm not familiar with that one, but maybe it is similar to cases where people look at whether including a training data point affects the results. Steinke's work (e.g., Privacy Auditing with One (1) Training Run) always has some similar setup where there is an indicator variable about whether we include a variable/sample.
- The fact that we go from a combinatorial problem to a tractable optimization is one of the coolest benefits, and it was nice to see how this smooths out the behavior. It reminds me a little of the situation with the Sinkhorn algorithm and entropy-regularized optimal transport. It idly makes me wonder if, instead of a relaxed bottleneck, there is a natural entropy-regularized version that would also work. (Just an idle comment, no idea how that might look.)
Author Response
We thank the referee for taking the time to review our manuscript and for their helpful comments. Below we respond point-by-point. Important changes in the manuscript have been marked in dark red.
> - It was surprising to see the joint distribution is not used! Is it fair to
> say, though, that even though only the conditional distribution p(y|x_s) is
> used, this somewhat constrains what the joint distribution p(x1, ..., xn)
> could be? I'm not very familiar with the PID corner cases, but are there
> any that depend on the joint distribution p(x1, ..., xn)?
In general, the Blackwell redundancy does not depend on the joint distribution, although of course it is true that only some joint distributions can be compatible with a given set of pairwise marginals . Several other previously proposed measures of redundancy also do not depend on the full joint distribution, only the target marginal p(y) and the channels p(x_s|y), and this property was even proposed as a basic requirement of any redundancy measure in Bertschinger et al, 2014. We provide a list of previous redundancy measures and whether they satisfy this property or not in Table 1 of our previous paper (Kolchinsky 2022). We have expanded a paragraph at the end of Section II to clarify this issue.
>-One part that was a little unclear to me is how the domain of Z depends
> on label identities. For instance, if I say that X1 is in {0,1} and X2 is in
> {2,3,4}, does it change things? I'm trying to understand how general this
> part is. My guess is that since Q depends on S and Z, it doesn't matter at
> all, and the domain union step was just a notational choice. But it would
> be nice to clarify.
This is a good point to clarify. The set of joint outcomes of S and Z with non-zero probability — on which Q depends — is the disjoint union of the outcomes of the individual sources. For this reason, our results do not depend on the precise labeling of the sources. We have added a paragraph before Theorem 1 to clarify this.
> - Is there some natural way to set the measure ν(s)? I could imagine some
> sort of optimization to set it. Though the choice of a uniform distribution
> is simple and natural.
We appreciate the suggestion, and we have added a paragraph to the end of section III.C to clarify this point. We now discuss some examples of where v(s) might not be uniform, for instance if it represents the proportion of resources (like metabolism or volume) allocated to different sources, or the proportion of times that a source appears in the dataset. We also mention the possibility of an “adversarial” distribution where v(s) is chosen to maximize the redundancy bottleneck value.
> - In Eq. 10, there's a typo—missing a variable z.
Thank you, this has been corrected.
> - You mentioned the privacy funnel. I'm not familiar with that one, but
> maybe it is similar to cases where people look at whether including a
> training data point affects the results. Steinke's work (e.g., Privacy
> Auditing with One (1) Training Run) always has some similar setup where
> there is an indicator variable about whether we include a variable/sample.
We thank the reviewer for the pointer. From what we saw, in Steinke there is an indicator variable per individual data sample. This is different from our approach, where we are interested in compressing information about which variable (i.e., feature) provided the information.
> - The fact that we go from a combinatorial problem to a tractable
> optimization is one of the coolest benefits, and it was nice to see how this
> smooths out the behavior. It reminds me a little of the situation with the
> Sinkhorn algorithm and entropy-regularized optimal transport. It idly
> makes me wonder if, instead of a relaxed bottleneck, there is a natural
> entropy-regularized version that would also work. (Just an idle comment,
> no idea how that might look.)
This is an interesting comment. There is indeed some conceptual similarity to Sinkhorn but also some important differences. Specifically, the optimal transport problem (as appears in Sinkhorn) is convex but not strictly so (linear programming). Thus, even a small entropic regularization makes the problem strictly convex and allows qualitatively different — and much faster — algorithms to be used. This is different from our case, where the Blackwell redundancy objective is not at all convex (it’s actually strictly concave), and it does not become strictly convex after some small amount of entropic regularization. To summarize, the comparison with entropic regularization is suggestive, but the impact, it seems to us, is not sufficiently clear cut as to make a definitive statement in the manuscript.
Reviewer 2 Report
Comments and Suggestions for Authors
The submission proposes an information bottleneck scheme to extract the redundant bits of information from a set of sources about a target. Redundancy is defined as Blackwell redundancy, building off the author’s prior work on PID formulated around this specification of redundancy. The IB scheme is based on compressing the outcome of one randomly selected source at a time such that the information revealed about the source’s identity is constrained, while maximizing information about the target variable. By varying the magnitude of the constraint, a spectrum of the redundant bits of information is defined. One end of the spectrum is the redundant information contained in every source and the other end is the average information you can get from any one source. Between these extremes, information that is redundant to subsets of source variables can be identified.
An important contribution is an iterative algorithm similar to Blahut-Arimoto that is introduced to optimize the IB spectrum. Another interesting contribution is a decomposition of the compression and prediction terms into contributions from each source. In a handful of worked examples, the full curve is optimized and the source contributions are found. Finally some important properties of the redundancy bottleneck are established, such as that it leads to a smooth optimization problem even though Blackwell redundancy can have discontinuities.
I think the work is very interesting and certainly worth publishing. In addition to the value in the theory realm of a connection between PID and IB, I can see practical value where the redundancy bottleneck leads to unique insights about real world datasets. Namely, the ability to find redundancy across a range of compression could be revealing about the nature of the sources in relation to the target.
The examples are helpful and the writing is clear. Before publishing,
1. I am confused by the source contributions to the compression term. In Example 3, how can I understand the contribution of source 3 dropping to zero while maintaining a positive contribution to the prediction term? If the compression contribution is zero, its weighted deficiency must also be zero by the upper bound in the equation after (21), correct? That suggests I can obtain the compression Q (which should start to resemble sources 1 and 2 after the transition) from source 3, which seems incorrect.
Merely a suggestion: it might be more effective to just visualize the prediction contributions on the right vertical axes of the macro RB curve plots (panel c of each figure) rather than having an RB curve per source.
2. The title “PID as IB” seems to me overreaching and unnecessarily broad. The title suggests PID has been cast in terms of an IB setup, when only part (the pure redundancy terms) is meant to be identified by this setup. I would be happy to be rebutted on this point, or to see a title that is more specific and more reflective of the many valuable contributions in this work.
Author Response
We thank the referee for taking the time to review our manuscript and for their helpful comments. Below we respond to the two points raised by the referee. Important changes in the manuscript have been marked in dark red.
> 1. I am confused by the source contributions to the compression term. In
> Example 3, how can I understand the contribution of source 3 dropping to zero
> while maintaining a positive contribution to the prediction term? If the
> compression contribution is zero, its weighted deficiency must also be zero by
> the upper bound in the equation after (21), correct? That suggests I can
> obtain the compression Q (which should start to resemble sources 1 and 2 after
> the transition) from source 3, which seems incorrect.
We appreciate that this example is a bit counterintuitive. The referee is correct that as the compression contribution from source 3 decreases, the weighted deficiency between P(x_3|y) and P(q|y) must also decrease. However, even though prediction contributions increase from source 1 and 2 after the transition, it is not necessarily the case that P(q|y) must start to resemble those sources 1 and 2 after the transition. This is because in our formulation, prediction is quantified via the conditional mutual information I(Q;Y|S), not I(Q;Y). Therefore, prediction can increase due to changes in the joint distribution p(q,y,s), even while the pairwise marginal p(q,y) achieves a lower deficiency with respect to source 3.
More generally, we emphasize that it is possible for the prediction contribution from a given source to stay the same, or even increase, even while the compression cost contributed by that source decreases. In other words, the RB curves for individual sources may not be monotonic (see Figure 2(d)). However, the overall RB curve (Figure 2(c)), which reflects the tradeoff between average prediction/compression contributions from all sources, must always be concave and monotonic.
We have added two paragraphs when discussing Example 3 to explain this.
> Merely a suggestion: it might be more effective to just visualize the prediction
> contributions on the right vertical axes of the macro RB curve plots (panel c
> of each figure) rather than having an RB curve per source.
We appreciate the referee’s suggestion. We have done a slightly different modification: in subplot (a) and (b) of each figure, we decompose the overall compression and prediction terms into contributions from individual sources at various β.
> 2. The title “PID as IB” seems to me overreaching and unnecessarily broad. The
> title suggests PID has been cast in terms of an IB setup, when only part
> (the pure redundancy terms) is meant to be identified by this setup. I would
> be happy to be rebutted on this point, or to see a title that is more specific
> and more reflective of the many valuable contributions in this work.
This title was meant to suggest a general research direction that potentially also interpret other aspects and measures of the PID as IB-type compression/prediction tradeoffs. We agree with the reviewer, however, that this goes beyond what is actually presented in the manuscript. We have changed the title to “Partial information decomposition: redundancy as information bottleneck”.